# Incorporation of Mixing Microbial Induced Calcite Precipitation (MICP) with Pretreatment Procedure for Road Soil Subgrade Stabilization

**DOI:** 10.3390/ma15196529

**Published:** 2022-09-21

**Authors:** Xiaodi Hu, Xiongzheng Fu, Pan Pan, Lirong Lin, Yihan Sun

**Affiliations:** 1School of Civil Engineering and Architecture, Wuhan Institute of Technology, Wuhan 430205, China; 2Hubei Provincial Engineering Research Center for Green Civil Engineering Materials and Structures, Wuhan 430073, China; 3Hubei Provincial Academy of Building Research and Design Co., Ltd., Wuhan 430071, China; 4Zhejiang Academy of Transportation Sciences, Hangzhou 310023, China

**Keywords:** microbial induced calcite precipitation (MICP), pretreatment-mixing MICP method, subgrade soil stabilization

## Abstract

Microbial induced carbonate precipitation (MICP) provides an alternative method to stabilize the soil. To further improve the reinforcement effect, this study aims to propose a strategy by incorporating the mixing MICP method with pretreatment procedure. A series of laboratory tests were performed to investigate the preparation parameters (including the moisture content and dry density of the soil, the concentration of urea and CaCl_2_ in cementation solution), the engineering properties, the CaCO_3_ distribution as well as the mineralogical and micro structural characteristics of pretreatment-mixing MICP reinforced soil (PMMRS). Based on the orthogonal experiment results, the optimum preparation parameters for PMMRS were determined. The UCS of PMMRS was more strongly dependent on the moisture content and concentration of CaCl_2_ than the concentration ratio of CaCl_2_ to urea. Moreover, it was testified that incorporation of pretreatment procedure improved the stabilization effect of traditional mixing MICP method on the clayed sand (CLS). The UCS of PMMRS specimen was increased by 198% and 78% for the pure CLS and the simple mixing MICP reinforced soil, respectively. Furthermore, the CaCO_3_ products generated consisted of the aragonite, calcite and vaterite, which distributed unevenly inside the specimen no matter the lateral or vertical direction. The reason for the uneven distribution might be that oxygen content varied with the regions in different directions, and hence affected the mineralization reaction. In addition, the mineralization reaction would affect the pore structure of the soil, which was highly related to the stabilization effect of MICP reinforced soil.

## 1. Introduction

With the rapid development of economy and society, there has been an increasing demand for road construction in China during the past decades. By the end of 2020, the total road mileage reached 5,198,100 km. Since China is a vast country, the geological conditions might vary significantly between regions. For some projects, it is usually encountered that the engineering properties of soil cannot meet the specification requirements. In order to ensure the safety and durability of road structures, effective methods are needed to eliminate the potential risk caused by the poor soil character.

Generally, there are four common techniques to reinforce the road soil subgrade, including: (a) using natural or synthetic materials, e.g., geosynthetics, fibers, polyester, etc. [1,2]; (b) developing physical engineering methods, e.g., replacing the in situ poor soil, dynamic compaction, using sand/stone columns, etc. [3,4]; (c) injecting/grouting chemical materials, e.g., micro-fine cement, epoxy, phenoplasts, silicates, polyurethane, etc. [5,6]; (d) producing inorganic binder stabilization, e.g., Portland cement, lime, coal ash, etc. [7,8,9]. Among these methods, replacing the in situ poor soil and cement binder stabilization are the most common in China, but have considerable disadvantages that need to be seriously considered. On the one hand, replacing the in situ soil method is expensive and inefficient, although it seems to be the simplest. The alternative soil that meets the technical specification requirement needs to be explored and transported from somewhere else, while the replaced poor soil is needed to be removed and requires a large amount of land for disposal. It will bring considerable negative effects to the ecological environment as well. On the other hand, the cement binder stabilization is not environmentally friendly since cement production consumes a large amount of energy and significantly increases CO_2_ emissions [10,11,12,13]. Therefore, it is urgent to develop a more environmentally friendly and efficient method for stabilizing the poor soil subgrade.

Microbial Induced Calcite Precipitation (MICP), which is an emerging and eco-friendly technique based on microbial mineralization, has been developed and applied in soil reinforcement [14,15], permeability reduction [16,17], slopes stabilization [18,19], surface reparation [20,21], liquefaction resistance enhancement [22] and as a heavy metal pollution treatment [23]. According to the reaction principles, microbial mineralization process of the MICP can be divided into several categorizes, e.g., urea hydrolysis, denitrification [24], iron reduction, enzymatic-induced carbonate precipitation (EICP) [25]. Among them, the urea hydrolysis MICP has gained the most attention due to the quickly controllable reaction, high chemical conversion efficiency and excellent environmental tolerance. Moreover, *Sporosarcina pasteurii*, which is widely found in natural soil, has high urease activity and has becomes one of the most popular MICP bacteria species nowadays [26]. The main chemical reactions of urea hydrolysis MICP are summarized as Equations (1)–(6) [27,28]. Moreover, the mineralization reaction of *S. pasteurii* is often considered as being catalyzed by the carbonic anhydrase and urease. The urease could hydrolyze the urea into ammonia and carbon dioxide, thus rapidly increasing the pH value and carbonate concentration of the cell microenvironment (Equations (1)–(3)). Moreover, the carbonic anhydrase could catalyze the hydration reaction of carbon dioxide, forming carbonate and bicarbonate (Equations (4) and (5)), thus generating CaCO_3_ (Equation (6)) [28].
CO(NH_2_)_2_ + H_2_O → NH_3_ + NH_2_COOH(1)
NH_2_COOH + H_2_O → H_2_CO_3_ + NH_3_(2)
NH_3_ + H_2_O ↔ NH_4_^+^ + OH^–^(3)
H_2_CO_3_ ↔ HCO_3_^−^ + H^+^
(4)
HCO_3_^−^ + H^+^ + 2OH^−^ ↔ CO_3_^2−^ + 2H_2_O(5)
Ca^2 +^ + CO_3_^−^ + HCO_3_^−^ + OH^−^ → 2CaCO_3_ + H_2_O(6)

For reinforcing soil, the application of MICP can be divided into four categories, including the injecting [29] or the pre-injecting method [30], the spraying method [31], the soaking method [32,33] and the mixing method [34,35]. For the injecting/pre-injecting, spraying and soaking methods, the bacteria and cement substances need to be diffused from the surface to inside or from one side to the opposite. In this case, the CaCO_3_ generated by the mineralization reaction might block the pore throats and prevent the microorganisms bacteria and cement substances from diffusing, which would attenuate the reinforcement effect of MICP method [36,37]. From this point of view, these methods are only suitable for enhancing the coarse-grained soils, but have limited reinforcement effect on the fine-grained soils with dense structure. Compared to these methods, mixing MICP can lead soil particles contacting well with the bacteria and cement substances, and hence shows considerable advantages in treating both coarse-grained and fine-grained soils. Moreover, mixing MICP requires fewer bacteria, less cement substances and shorter treatment period. Therefore, it is recognized that the mixing method is more prospective for large-scale soil treatment, such as road soil subgrade, mine backfilling and so on.

As known, the maximal dry density and optimum moisture content are the critical parameters for soil compaction. The optimum moisture content of soil would limit the amount of solution containing the microorganisms and cement substances, which determines the reinforcement effect of mixing MICP method. Moreover, it is hard to prepare the solution with high concentration and high bacterial activity during the mixing process as well. On the other hand, the MICP process would consume a considerable amount of calcium. However, high concentration of calcium ions in solution would degrade the urease activity and thus attenuate the chemical conversion efficiency of MICP [14]. Consequently, the combination effects might lower the production of CaCO_3_ and minimize the reinforcement effect of mixing MICP method. Therefore, it is necessary to modify the traditional mixing MICP method for further application in road soil treatment.

This study aims to propose a strategy by incorporating the mixing MICP method with pretreatment procedure, which is expected to add more substances involved in the mineralization reaction without increasing the concentration of the cementation solution and bacterial solution. Firstly, an orthogonal experimentation was employed to obtain the optimum parameters for preparing the pretreatment-mixing MICP reinforced soil (PMMRS). Next, the engineering performance of PMMRS was evaluated by the unconfined compressive strength (UCS) test and the unconsolidated-undrained triaxial (UU) test, while the content and distribution of CaCO_3_ generated were analyzed by the acid wash method. Moreover, the PMMRS was studied by the field emission scanning electron microscope (FESEM), X-Ray Diffraction (XRD) and mercury injection apparatus (MIA) as well.

## 2. Materials and Methods

### 2.1. Soil

In this study, the soil, which was a kind of clayey sand (CLS), was obtained from an engineering project in Hong’an City, China. The particles less than 0.075 mm were analyzed by a laser particle sizer (Malvern Mastersizer 2000), while the particles greater than 0.075 mm were analyzed by the water washing method according to the Chinese Specification JTG 3430-2020. (Figure 1) The physical properties of the CLS were tested according to the JTG 3430-2020 and the results are shown in Table 1.

### 2.2. Bacteria Culture for MICP

The *S. pasteurii* was obtained from the China General Microbiological Culture Collection Center (CGMCCC) with strain numbers of ATCC-11859 and CGMCC 1.3687. The liquid culture medium was augmented with 20 g/L urea, 20 g/L tryptone, 5 g/L soy peptone, 5 g/L NaCl and 24 mg/L NiCl_2_ while the pH value of liquid was controlled as 8.0. After the sterile and inoculation treatment, the bacteria-bearing liquid culture medium was cultured in an oscillating incubator at 30 °C with a vibration rate of 180 rpm for 24 h. The number of microorganisms in the bacterial solution was measured by the 721 G visible spectrophotometer with a wavelength of 600 nm, and the OD600 of bacteria solution was kept at about 1.2.

### 2.3. Cementation Solution and Water

The CaCl_2_ and CO(NH_2_)_2_ were used to prepare the cementation solution. The CaCl_2_ has five concentration levels (1.0, 2.0, 3.0, 4.0 and 5.0 mol/L) and the concentration ratios of CaCl_2_ to CO(NH_2_)_2_ were 1:1, 1:2, 1:3, 2:1 and 3:1, respectively. The water used to prepare the liquid culture medium and cementation solution was the deionized water, and the rest was tap water unless otherwise noted.

### 2.4. Sample Preparation

#### 2.4.1. Soil Pretreatment

In this study, the CLS was pretreated with the cementation solution by three steps. Firstly, the amount of cementation solution was determined based on the mass and moisture content of the CLS. Secondly, the CLS was mixed uniformly with the cementation solution for 5 min. Finally, the mixed CLS was dried at 105 °C until it achieved a constant weight.

#### 2.4.2. Specimen Preparation and Curing

For the UCS test, cylindrical specimens with a diameter of 50 mm and a height of 50 mm were prepared. Figure 2 illustrates the preparation procedure for the UCS specimen. Firstly, the inner wall of the mold was coated with petroleum jelly to prevent the CLS from sticking to the mold. Secondly, the pretreated CLS was mixed with the bacteria solution and then filled in the mold. The CLS mixture was compacted until two blocks were pressed into both the sides of the mold. Finally, the compacted specimens were demoulded and then cured in a biological incubator at 30 °C within different curing times before further tests.

For the UU test, specimens with a diameter of 39.1 mm and a height of 80 mm were compacted with a hammer in four layers. Each layer was compacted well before adding CLS mixture for the next layer. The subsequent preparation and curing procedures were similar with the UCS specimen.

### 2.5. Test Method

#### 2.5.1. Unconfined Compressive Strength (UCS)

In this study, the UCS test was conducted with a loading rate of 1 mm/min by the universal testing machine (UTM-100). The sizes of the cured specimens were measured before test.

#### 2.5.2. Undrained–Unconsolidated (UU) Triaxial Test

The UU test was performed by the triaxial testing machine. Firstly, the cured specimens were vacuum saturated for 24 h before test. In this study, the specimen was axially loaded with a rate of 0.4 mm/min under three fixed pressure levels, including 100 kPa, 200 kPa and 300 kPa. The test was finished when the axial strain reached to 20%.

#### 2.5.3. Content and Distribution of CaCO_3_

The content and distribution of CaCO_3_ in the PMMRS specimens were investigated by the acid washing method [38]. In order to ensure the sampling representation, the samples were obtained from the nine locations of the specimen. Figure 3 illustrates the sampling strategy considering both the lateral and vertical directions.

Before testing, the sample was firstly washed by the deionized water to remove the soluble salts and then dried to a constant mass m_a_. Secondly, the dried sample was washed by the HCl solution several times until all the CaCO_3_ reacted with the HCl. Finally, the sample was washed by the deionized water again to remove the calcium and then dried to a constant mass m_b_. The content of CaCO_3_ (*CCC*) of sample can be calculated by Equation (7).
(7)$$CCC=(100-\frac{m_{b}}{m_{a}})\times 100\%$$

#### 2.5.4. X-ray Diffraction (XRD)

The mineralogical composition of the pure CLS and PMMRS was measured by the X-ray diffractometer (D8 ADVANCE). The samples were dried at 105 °C for 12 h and then pulverized to pass through the 0.075 mm sieve. During the test, the scanning speed was 3° per minute and the scanning angle was from 10° to 80° in units of 2-theta.

#### 2.5.5. Field Emission Scanning Electron Microscope (FESEM)

A Zeiss GminiSEM 300 FFESEM was employed to analyze the morphology and distribution of CaCO_3_ in the CLS, which can be further used to investigate the bonding characteristics between the CLS particles. The samples obtained from the cylindrical specimens had their surfaces smoothed before they were then dried at 105 °C for 12 h before testing.

#### 2.5.6. Mercury Injection Apparatus (MIA)

The particle and pore structure characteristics inside the specimens were investigated by a micromeritics AutoPore Iv 9520 (Montgomeryville, America, MicroActive AutoPore V 9600 2.03.00) mercury injection apparatus. Cubic samples with a length of 15 mm were taken from the specimens. All the samples were smoothed and dried at 105 °C for 12 h before test.

### 2.6. Orthogonal Experiment Design

In this study, an orthogonal experimentation was used to investigate the different factors that influence the reinforcement effect of pretreatment-mixing MICP method, including the maximal dry density and optimum moisture content of the CLS, the calcium ions concentration and the ratio of CaCl_2_ to urea in the cementation solution. Since the dry density of CLS is quantitatively determined by the moisture content, the moisture content can be considered as a factor that represents the dry density. Therefore, the orthogonal experimental was designed with three parameters and five levels, as in Table 2. The unconfined compressive strength was adopted as the evaluation indicator for the orthogonal experiment. Before the UCS test, the samples for each group were cured at 30 °C for 7 days and then dried at 105 °C for 24 h. To ensure the accuracy, six duplicate samples were prepared for each group and the corresponding average values were used for the statistical analysis.

## 3. Results and Discussion

### 3.1. Orthogonal Experiment Results

The specimens for different PMMRS groups were tested and the corresponding average UCS is summarized in Table 3. The combination effects of moisture content, concentration of CaCl_2_ and the ratio of CaCl_2_ to urea have a considerable effect on the UCS of PMMRS. For example, the average UCS of specimens for the No. 18 group (moisture content 9%, concentration of CaCl_2_ 3 mol/L and the ratio of CaCl_2_ to urea 1:2) was 12.92 MPa, which was 5.2 times more than the No. 15 group (moisture content 11%, concentration of CaCl_2_ 3 mol/L and the ratio of CaCl_2_ to urea 2:1).

The range analysis was used to determine the optimum parameters for the pretreating-mixing MICP process, and the results are shown in Table 4. The K_i_ is the sum of the average UCS results for the level i in each column. Since UCS represents the engineering property of PMMRS, the level corresponding to the maximum K_i_ in each column was adopted to determine the optimum parameters for preparing the PMMRS specimen. Therefore, the moisture content was determined as 9%, the concentration of CaCl_2_ 3.0 mol/L and the concentration ratio of CaCl_2_ to urea 1:1. In addition, when the moisture was 9% by weight of the CLS, the corresponding maximum dry density was 2.09 g/cm^3^.

As shown in Table 4, the UCS of PMMRS decreased with the moisture content. When the moisture content increased from 9% to 13%, the dry density of PMMRS decreased from 2.09 g/cm^3^ to 1.98 g/cm^3^, leading to the strength of PMMRS decreasing. Since high concentration of calcium ions would degrade the activity of urease, the UCS of PMMRS increased initially with the concentration of CaCl_2_ in the cementation solution and then showed a decrement trend when the CaCl_2_ concentration was greater than 3 mol/L.

Furthermore, the range value R was used to evaluate the effect significance of the preparation parameters on the UCS, which was calculated by Equation (8). The greater the R value, the more significant the parameter. As shown in Table 4, the R value for the moisture content was close to the concentration of CaCl_2_, which were obviously greater than the ratio of CaCl_2_ to urea. It indicated that the moisture content and the concentration of CaCl_2_ showed similar effect significance on the UCS of PMMRS, while the concentration ratio of CaCl_2_ to urea showed the lowest significance.
R = max{K_1_, K_2_, K_3_, K_4_, K_5_} − min{K_1_, K_2_, K_3_, K_4_, K_5_}(8)

Based on the test results, variance analysis was also conducted to quantitatively investigate the effect significance of parameters of MICP process on the UCS of PMMRS, and the results are shown in Table 5. Since each parameter has five different levels, the degree of freedom was determined as four. Generally, the greater F-Value indicates that the parameter shows more significance for the evaluation index. The significance order of the parameters was the moisture content of the CLS (dry density), the concentration of CaCl_2_ and the concentration ratio of CaCl_2_ to urea in cementation solution. The results demonstrated that moisture content and concentration of CaCl_2_ showed a more significant effect on the UCS of PMMRS than the concentration ratio of CaCl_2_ to urea, which was in accordance with the results in Table 4. 

In order to validate the effectiveness of orthogonal experiment, UCS test was performed on an additional groups of specimens, which were prepared according to the optimum parameters determined by the orthogonal experiment. The average UCS strength for the additional groups was 15.92 MPa, which was higher than all orthogonal experiment groups. It testified that the optimum parameters obtained by the orthogonal experiment can be used to prepare PMMRS with the greatest UCS strength.

### 3.2. Stress-Strain Characteristics of PMMRS in UCS Test

To study the mechanical characteristic of PMMRS, UCS test was performed on the PMMRS and simple mixing MICP reinforced soil (SMMRS) specimens. The PMMRS specimens were prepared according to the optimum parameters obtained by the orthogonal experiment. Additionally, the pure CLS, the bacteria solution and the cementation solution were used to prepare the SMMRS specimen. Additionally, the SMMRS group’s solutions have half the volume, respectively, and the same concentration as the solutions in the PMMRS group. In addition, pure CLS specimens with 9% moisture content were also prepared and tested as the control group. During the UCS test process, both the stress applied to the specimens and the corresponding strain was recorded (Figure 4).

The average UCS for the PMMRS group was the greatest, which was 15.92 MPa, while the corresponding results for the SMMRS and pure CLS specimens were 8.92 MPa and 4.72 MPa, respectively. The average UCS of the PMMRS group was almost three times greater than the pure CLS and was 78% greater than the SMMRS group. Moreover, the displacements under the peak stress for the PMMRS specimens were greater than the pure CLS and SMMRS. Results implied that the pretreatment-mixing MICP method exhibit a higher improvement over the simple mixing MICP method.

### 3.3. UU Test

Figure 5 presents the shear response for the PMMRS, SMMRS and pure CLS specimens under a fixed stress of 100 kPa. For the UU test, the PMMRS specimens were prepared with the optimum parameters determined by the orthogonal experiment, and the pure CLS specimens were prepared with the same moisture content of 9%. As shown in Figure 5, the deviatoric stress for the MICP reinforced specimen was obviously greater than the pure CLS at the same axial strain, and the peak deviatoric stress increased with the increase in reinforcement effect. In addition, there were difference in the shear responses among the PMMRS, SMMRS and pure CLS specimens. For the PMMRS specimen, the deviatoric stress initially increased with the axial strain and then tended to decrease after reaching the peak deviatoric stress. From this point of view, the strain–shear pressure response of PMMRS could be considered as strain softening. Moreover, the corresponding response for the pure CLS was strain hardening. The deviatoric stress continually increased with the axial strain, and there was no peak stress appearing. For the SMMRS specimens, the strain–shear pressure response was between strain hardening and strain softening. The reason for this phenomenon might be the influence of MICP reinforcement effect. Montoya [39] found that the stress–strain response of specimens transitioned from strain hardening to strain softening with the MICP reinforcement effect. According to the results in Section 3.2, the MICP reinforcement effect of PMMRS was superior to SMMRS. Therefore, the stress–strain response of PMMRS was strain hardening, and the response of SMMRS was between strain hardening and strain softening.

The Shear strength envelope for the PMMRS, SMMRS and pure CLS were obtained (Figure 6). The friction angle for the PMMRS, SMMRS and pure CLS specimens were 9.38°, 9.25° and 8.55°, while the cohesion values were 33.98 kPa, 21.57 kPa and 14.54 kPa, respectively. As published by Mingjuan Cui [40] and Ivanov Volodymyr [41], the reinforcement mechanism of MICP on the soil can be summarized as two types of principles: bio-clogging and bio-cementation. On the one hand, the CaCO_3_ crystals attach on the soil particles and improve the surface roughness, which increases the friction angle of the soil. On the other side, the CaCO_3_ crystals may bond soil particles together and hence improve the cohesion of the soil. Compared with the pure CLS, the cohesion of PMMRS and SMMRS showed a significant improvement while the friction angle increased by only 0.83° and 0.7°, respectively. It can be concluded that the bio-cementation principle mainly contributes to reinforcement effect for the mixing MICP method. Furthermore, both the friction angle and cohesion of PMMRS were greater than SMMRS, which also indicated that the pretreatment procedure could improve the reinforcement effect of the mixing MICP method.

### 3.4. CaCO_3_ Distribution Characteristics

There is no doubt that the reinforcement effect of MICP on the soil is strongly dependent on the amount and distribution of CaCO_3_ generated. In fact, it is really a great challenge that CaCO_3_ is generated unevenly in the specimen, no matter the laboratory experiment or the field test [14]. Figure 7 presents the CaCO_3_ content (*CCC*) test results of PMMRS and SMMRS in different sampling locations, which was described in Section 2.5.3. The *CCC* of PMMRS was greater than SMMPRS, while there was obvious difference in the *CCC* for different locations of both PMMRS and SMMRS specimens.

The average *CCC* for the specimen was used to study the relationship between the *CCC* in specimens and the CaCl_2_ concentration in cementation solution. The calcium ions conversion rate was adopted to evaluate the efficiency of MICP process, which can be calculated by Equations (9) and (10). The results of average *CCC* and calcium ions conversion rate (*CICR*) in different specimens are shown in Figure 8.
(9)$$CCC_{max}=\frac{m_{a}\times \frac{9\%}{1000}\times 3.0\times 100.09}{m_{a}}\times 100\%$$
(10)$$CICR=\frac{CCC}{CCC_{max}}$$
where *CCC_max_* is the theoretical maximum *CCC* in specimen.

The average *CCC* of PMMRS was increased by 74% more than the SMMRS while SMMRS had a higher *CICR* (Figure 8). Compared to the simple mixing MICP method, pretreatment-mixing MICP method could provide two times calcium ions for mineralized bacteria to generate CaCO_3_ theoretically. Therefore, PMMRS could have higher *CCC* and better mechanical properties. However, excessive calcium ions could also decrease the activity of urease in bacteria solution and attenuates the efficiency for urea hydrolysis. Consequently, PMMRS would have lower *CICR*.

In this study, the distribution characteristics of CaCO_3_ in both lateral and vertical directions were studied as well. The *CCC* in different parts was calculated by averaging the *CCC* of sampling locations as the corresponding column or line in Table 6. For example, the *CCC* of the edge part was the average *CCC* of sampling locations 1, 4 and 7; while the *CCC* of the downside part was the average *CCC* of sampling locations 7, 8 and 9.

The results of CaCO_3_ distribution in the lateral and vertical directions are shown in Figure 9. CaCO_3_ distribution patterns for PMMRS and SMMRS were approximately similar. In the lateral direction, the *CCC* of the edge part was the greatest, followed by the central part and the middle parts. In the vertical direction, the *CCC* of the specimen showed a decrease trend from the top to the bottom. The reason might be attributed to the effect of oxygen on the mineralized bacteria. *S. pasteurii* is a kind of microaerobic microorganism, which requires oxygen to survive and reproduce. The corresponding mineralization reaction is considerably affected by the oxygen. Jain Surabhi [42] reported that the urease activity and the amount of CaCO_3_ formed by the *S. pasteurii* in the anaerobic condition is more negligible than the aerobic condition. In addition, the dissolved oxygen in solution could influence the UCS and the CaCO_3_ distribution in specimens [43]. For the *S. pasteurii*, there are three ways to obtain the oxygen in the PMMRS specimen, including the dissolved oxygen in solution, oxygen existing in the pore and the oxygen from outside environment. Generally, it is hard for the bacteria inside the specimens to obtain the oxygen from the environment if far away from the surface. Therefore, once the oxygen in solution and in the pores were consumed completely, the mineralized bacteria could no longer survive, reproduce and generate CaCO_3_. Consequently, *CCC* of edge part was the greatest in the lateral direction while *CCC* decreased from the top to the bottom in the vertical direction.

### 3.5. Curing Time

In order to study the strength development of PMMRS, specimens were prepared and cured in a microbial incubator at 30 °C for 1, 3, 5, 7, 14, 21 and 28 days, respectively. After the specimens were dried at 105 °C for 24 h, they were subjected to UCS testing. The UCS growth rate was defined and calculated by Equation (11). The UCS and UCS growth rate for the PMMRS specimens within different curing periods are shown in Figure 10.
(11)$$\mathrm{UCS}\;\mathrm{growth}\;\mathrm{rate}=\frac{UCS_{i}}{UCS_{28d}}\times 100\%$$
where *UCS_i_* and *UCS_28d_* are the average UCS of the PMMRS specimens cured for i days and 28 days, respectively, MPa.

UCS of PMMRS increased within the whole curing period (Figure 10). It implied that the alive mineralized microorganisms continually consumed the cementation substances and nutrients, and produced the CaCO_3_ to reinforce the CLS. Moreover, it was evident that the UCS development was strongly dependent on the curing period. After 1 day curing, the average UCS of PMMRS was 6.86 MPa and the UCS growth rate was approximately 43% to the corresponding results of specimens cured for 28 days. For the specimens cured for 3 days, the average UCS was increased almost 2 times more than the specimens cured for 1 day, and the corresponding UCS growth rate was 85%. In the rest curing period, the UCS increased steadily, but the UCS improvement was relatively small. From this point of view, 3 days could be determined as the curing time for PMMRS specimen before test.

### 3.6. XRD Test

Figure 11 presents the X-ray diffraction test results for the pure CLS and PMMRS. There were three kinds of mineral components in the pure CLS sample, including the quartz, albite and anorthite. Compare to the pure CLS sample, CaCO_3_ was generated by the MICP process to reinforce the CLS. However, it was notable that there were three types of CaCO_3_ in PMMRS sample, including vaterite, calcite and aragonite. Due to the difference in crystal structures, these CaCO_3_ products might show different reinforcement effect on the CLS. Therefore, further study is needed to determine the generation mechanisms and reinforcement effect of the CaCO_3_ products in PMMRS.

### 3.7. FESEM Test

Figure 12 and Figure 13 are the FESEM images for the pure CLS and the PMMRS specimens, respectively. For the pure CLS, small soil particles gathered and formed on the big particles in Figure 12a. As shown in Figure 12b, CaCO_3_ crystals were easily observed for the PMMRS specimen. Firstly, CaCO_3_ crystals could bond the little soil particles together and thus increases the cohesion of the CLS. Moreover, some CaCO_3_ crystals were attached to the surface of soil particles, which improves the internal friction between the soil particles. Therefore, the PMMRS specimens showed greater cohesion and friction angle than the pure CLS and hence contributes to greater strength, which was in accordance with the results of UCS and UU tests. Moreover, Figure 13 illustrates that three CaCO_3_ crystals, including calcite, aragonite and vaterite, were generated in PMMRS, which showed obvious differences in the morphology features. This was in accordance with the results in Section 3.6. The rules for CaCO_3_ generation and the effect mechanism on PMMRS are needed to be seriously clarified from the micro structural point of view in the future study.

### 3.8. MIA Test

The MIA test results are shown in Figure 14 and Figure 15. As shown in Figure 14, for the small pore size, the pore volumes of the PMMRS and SMMRS were less than the pure CLS, while the results for the large pore size were the opposite. Figure 15 presents that PMMSR had the highest average pore diameter but lowest calculated porosity, while pure CLS did the opposite. 

The variations of cumulative intrusion and average pore diameter could be attributed to the CaCO_3_ crystals generated in MICP reinforced soil. Due to the blocking effect of the CaCO_3_ crystals, the proportion of pores with a smaller size for all the MICP reinforced soil was smaller than the pure CLS. Since the small soil particles could be bonded together, the amount of little particles would decrease, leading the soil pores that are normally filled with these little particles to become unblocked. Therefore, the average pore diameter in MICP reinforced soil would be increased. Referring to the UCS results of Section 3.2, the UCS of specimens increased with the decrement of the calculated porosity as well as the increment of the average pore diameter. It can be concluded that the reinforcement effect of MICP reinforced soil was highly related to the pore structure of the soil.

## 4. Conclusions

The present work aimed to propose a strategy for mixing MICP method incorporated with the pretreatment procedure to stabilize the soil in road construction projects. A series of laboratory tests were performed to investigate the preparation parameters, the engineering properties, the distribution characteristics of CaCO_3_ as well as the mineralogy and micro structure of PMMRS. The key findings and recommendations drawn from the study are listed below:(1)The orthogonal experiment results determined the optimum parameters for preparing PMMRS, i.e., moisture content of 9%, the concentration of CaCl_2_ 3.0 mol/L, and the concentration ratio of CaCl_2_ to urea 1:1. The moisture content and the concentration of CaCl_2_ showed the significant effect on the UCS of PMMRS. In contrast, the effect of the concentration ratio of CaCl_2_ to urea was relatively low.(2)The pretreatment procedure could significantly improve the reinforcement effect of mixing MICP method on the CLS. Compared to the pure CLS and SMMRS specimen, the UCS of PMMRS specimen was increased by 198% and 78%, respectively. The reinforcement effect could be attributed to improvement of cohesion of the soil. Moreover, the mineralization reaction would affect the pore structure of the soil, which was highly related to the reinforcement effect of MICP reinforced soil.(3)After 3 days of curing, the UCS growth rate of the specimens increased rapidly to 85% of the specimens cured for 28 days. Therefore, 3 days was recommended as the curing time for PMMRS specimen before performing the test.(4)The CaCO_3_ distribution inside the specimen was not even no matter the lateral or vertical direction. The *CCC* of edge part was the greatest in the lateral direction while it decreased from the top to the bottom in the vertical direction. The reason might be attributed to the effect of oxygen on the mineralized bacteria. However, the strength of PMMRS was not monotonously positively related to the *CCC* with the increment of the CaCl_2_ concentration.

To facilitate the application of pretreatment-mixing MICP method, further research can be conducted into the generation mechanisms and reinforcement effect of different CaCO_3_ products in PMMRS. Moreover, it is strongly recommended to investigate the methodology for ensuring that the CaCO_3_ is generated more evenly in the PMMRS. Furthermore, long-term performance of PMMRS should also be investigated in order to verify the indoor test results in further studies.

## Figures and Tables

**Figure 1 materials-15-06529-f001:**
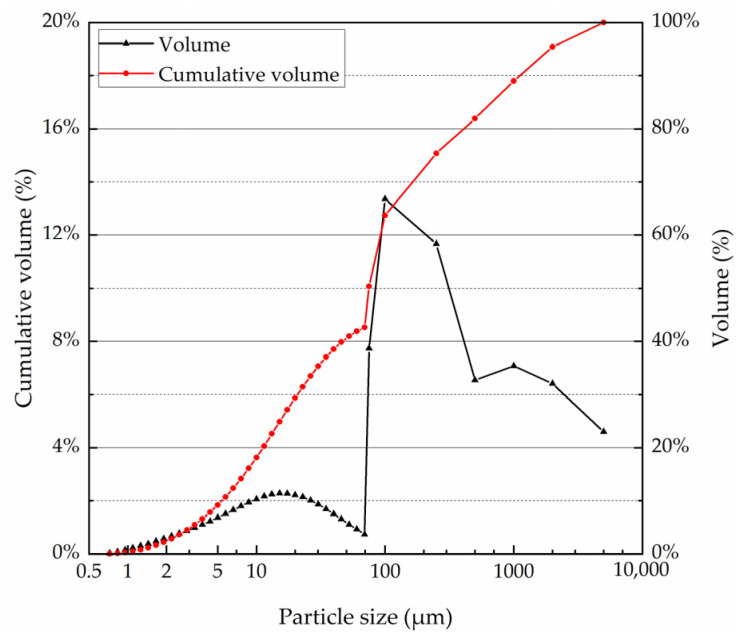
Particle size distribution of the CLS.

**Figure 2 materials-15-06529-f002:**
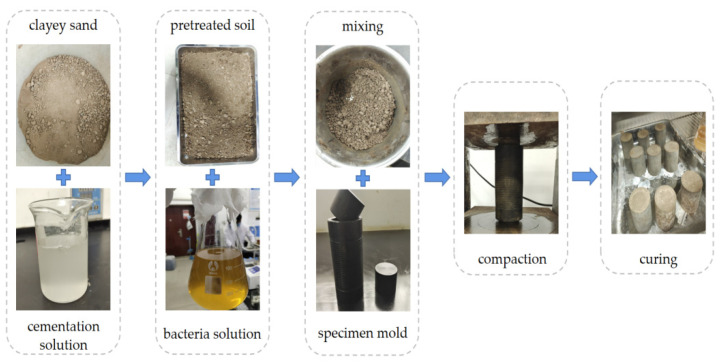
The specimen preparation of UCS test.

**Figure 3 materials-15-06529-f003:**
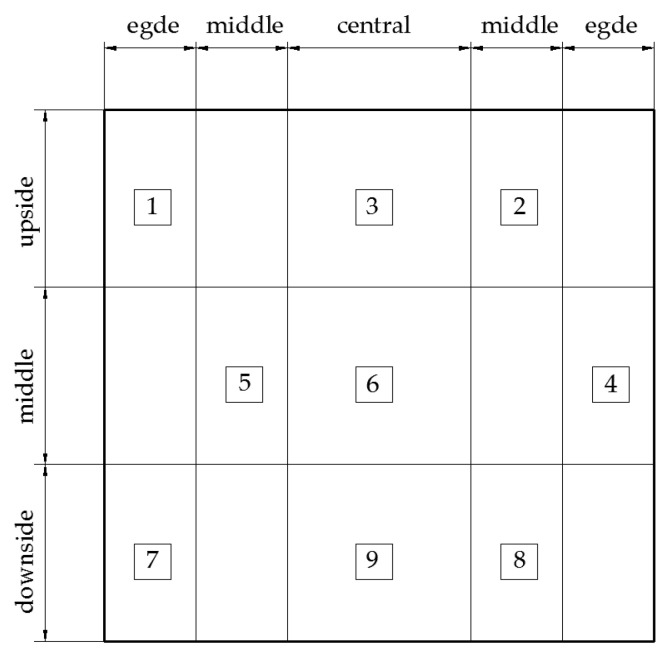
Strategy for obtaining the *CCC* sample.

**Figure 4 materials-15-06529-f004:**
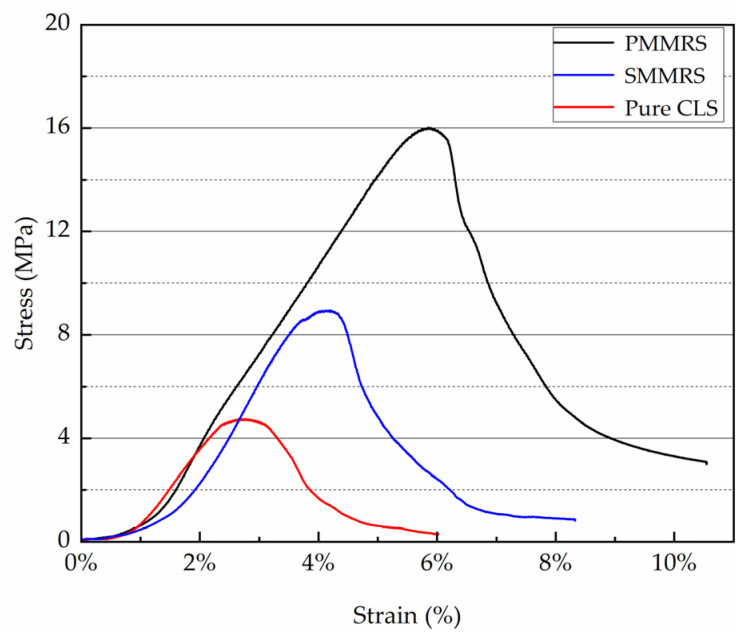
The strain–stress curve of PMMRS, SMMRS and pure CLS.

**Figure 5 materials-15-06529-f005:**
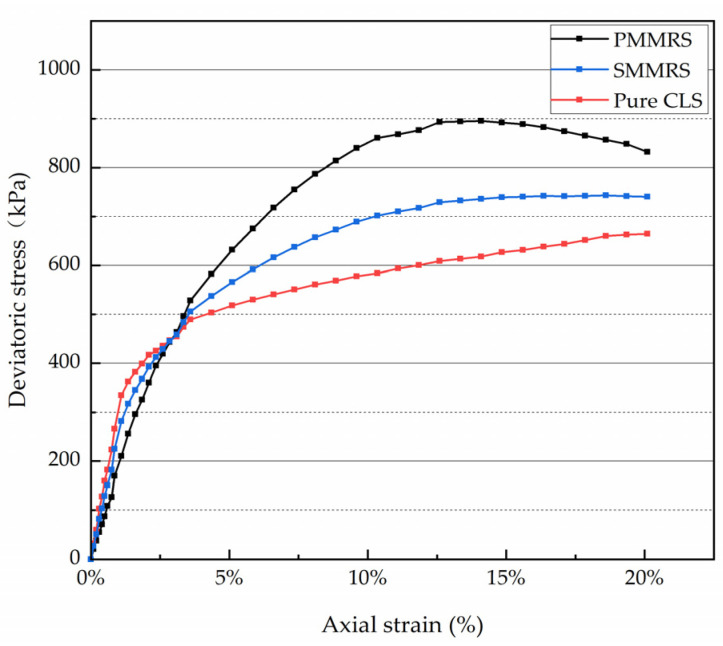
Shear response of specimens at confining pressure of 100 kPa.

**Figure 6 materials-15-06529-f006:**
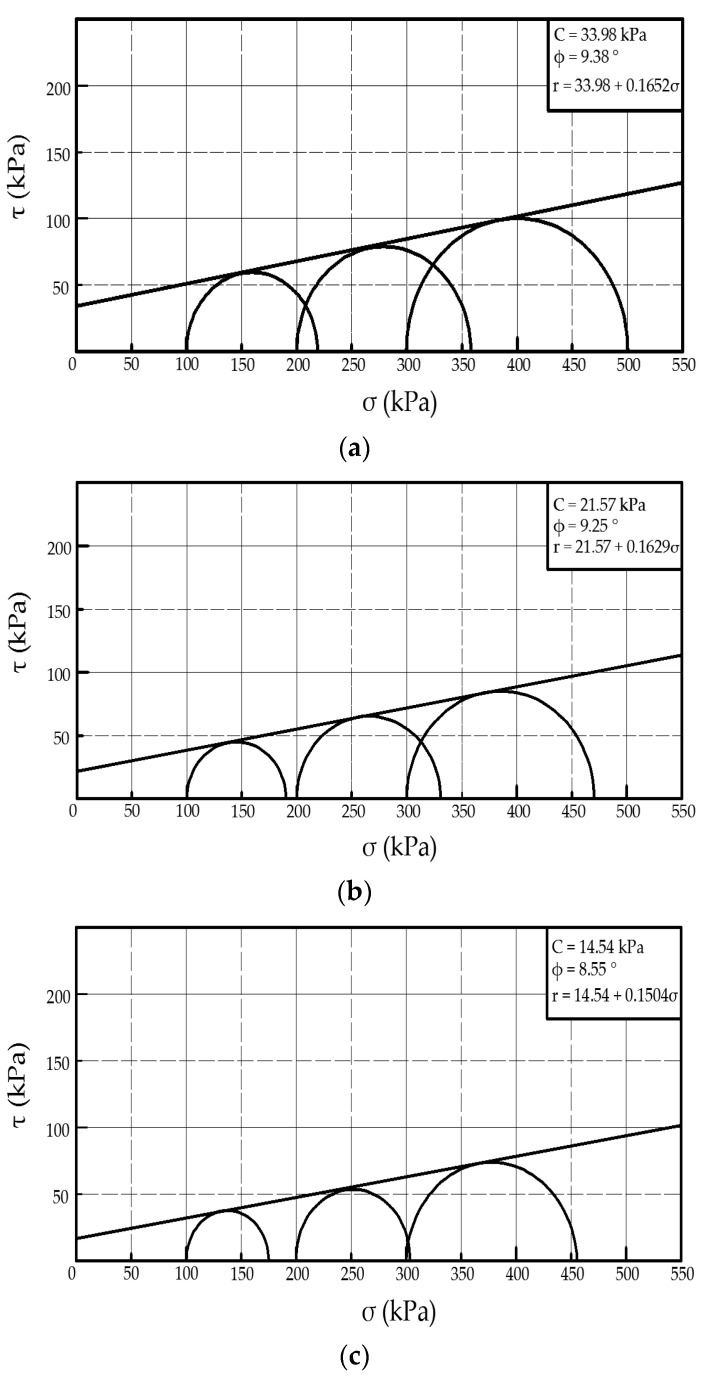
Shear strength envelope of UU test. (**a**) PMMRS, (**b**) SMMRS and (**c**) Pure CLS.

**Figure 7 materials-15-06529-f007:**
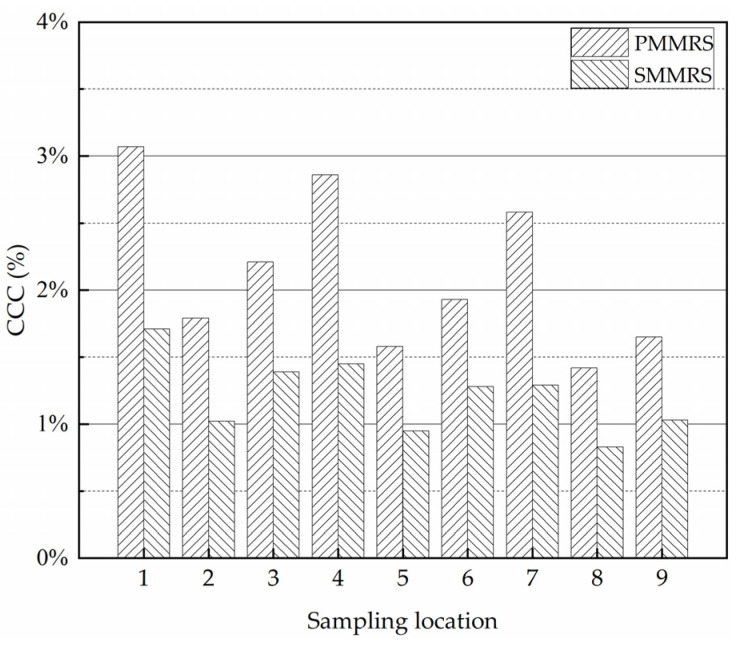
*CCC* test results of PMMRS and SMMRS in different sampling locations.

**Figure 8 materials-15-06529-f008:**
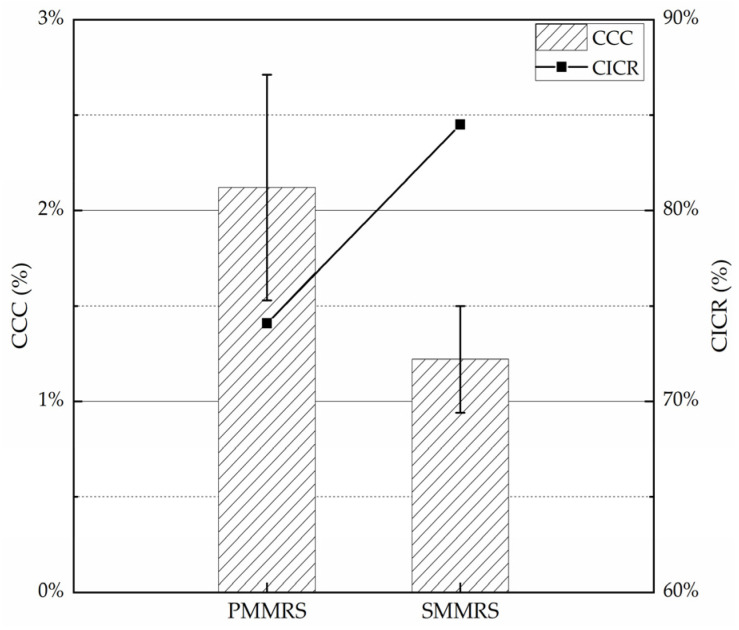
The average *CCC* and *CICR* of PMMRS and SMMRS.

**Figure 9 materials-15-06529-f009:**
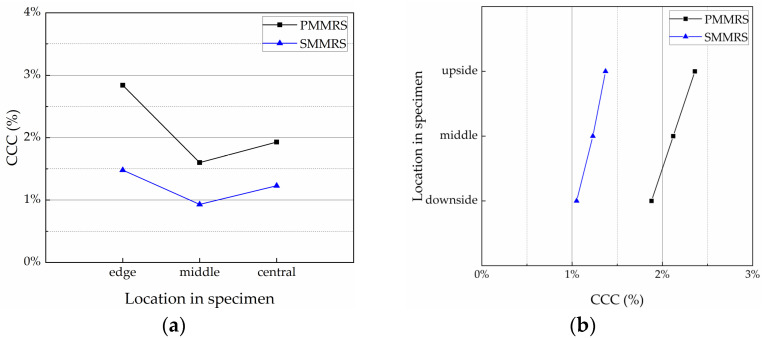
CaCO_3_ distribution in PMMRS and SMMRS: (**a**) lateral direction and (**b**) vertical direction.

**Figure 10 materials-15-06529-f010:**
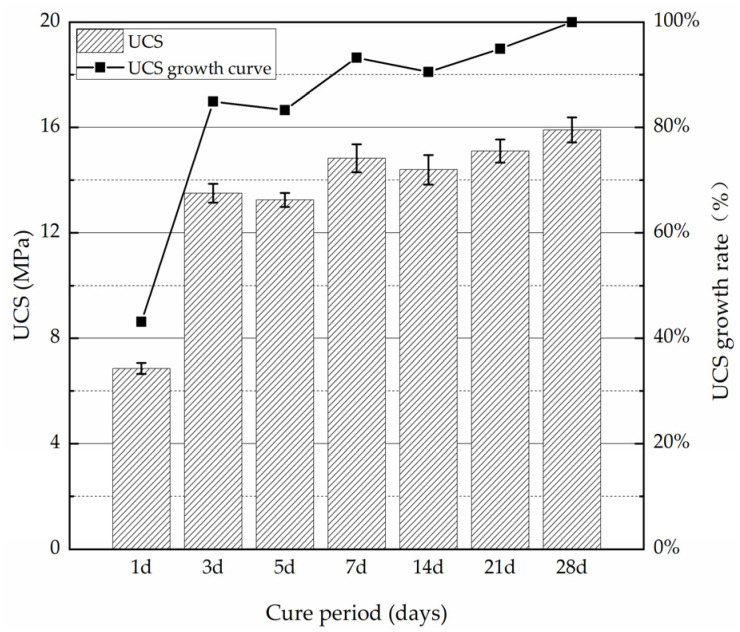
UCS development of PMMRS during the curing period.

**Figure 11 materials-15-06529-f011:**
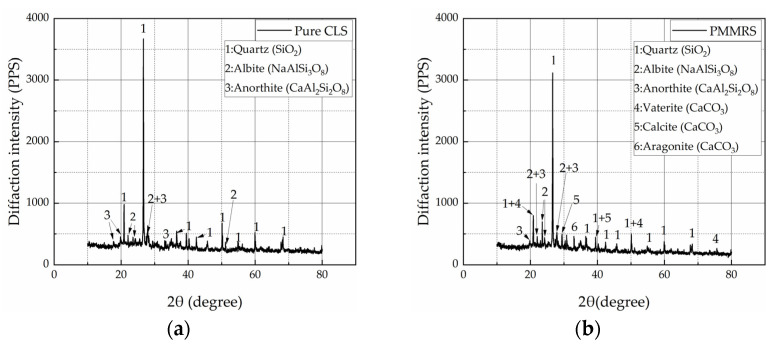
The XRD results of (**a**) pure CLS and (**b**) PMMRS.

**Figure 12 materials-15-06529-f012:**
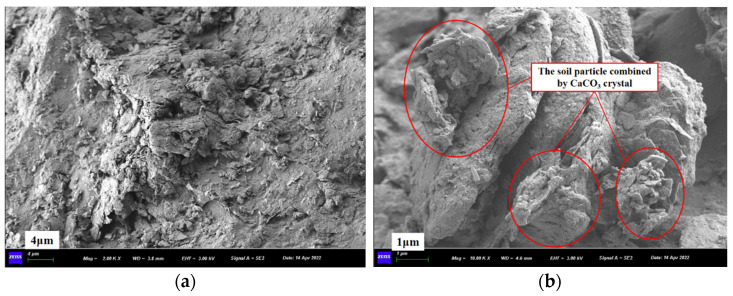
FESEM imagines of (**a**) pure CLS and (**b**) PMMRS.

**Figure 13 materials-15-06529-f013:**
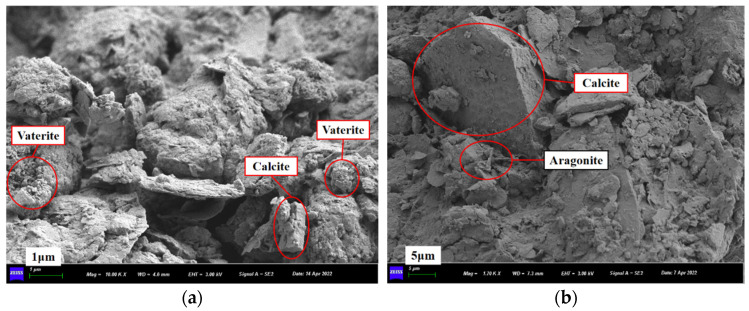
Different CaCO_3_ crystal morphology in PMMRS: (**a**) vaterite and calcite, (**b**) calcite and aragonite.

**Figure 14 materials-15-06529-f014:**
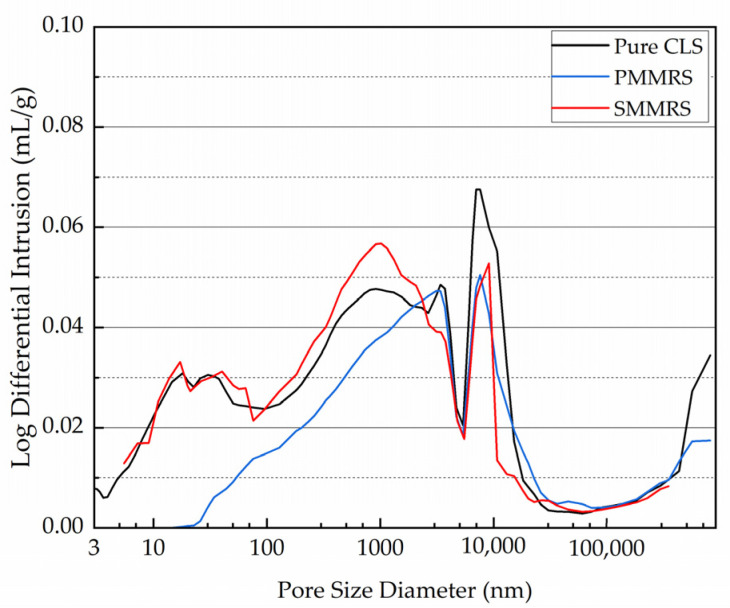
The pore distribution and volume of PMMRS, SMMRS and pure CLS.

**Figure 15 materials-15-06529-f015:**
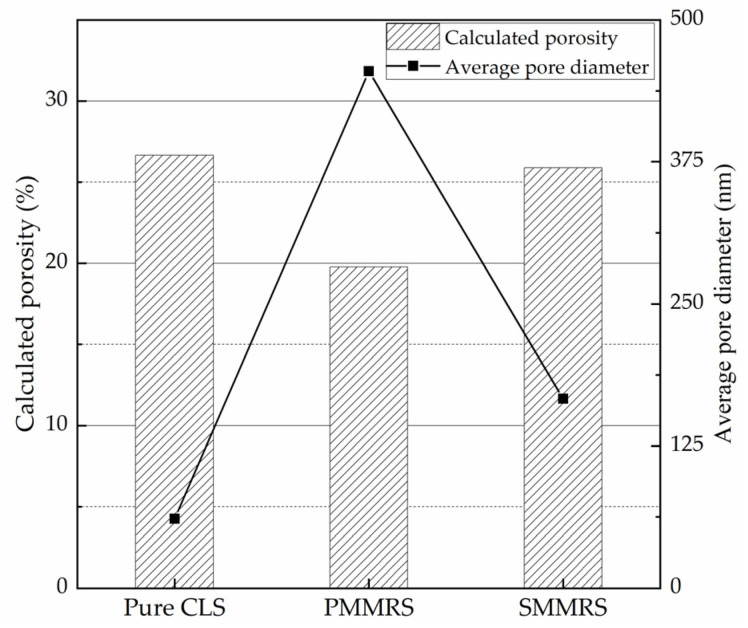
The calculated porosity and average pore diameter of PMMRS, SMMRS and pure CLS.

**Table 1 materials-15-06529-t001:** Physical properties of the CLS.

Indexes	Results	Indexes	Results
Classification	Clayey Sand (CLS)	Clay	2.20%
Liquid limit	46.52%	Silt	7.00%
Plastic limit	25.61%	Sand	86.20%
Plastic index	20.91	d_10_	0.0054 mm
Specific gravity	2.71	d_30_	0.0209 mm
Maximum dry density	2.09 g/cm^3^	d_60_	0.9801 mm
Optimum moisture content	9.2%	C_u_	181.3474
pH	6.87	C_s_	0.0821

**Table 2 materials-15-06529-t002:** The orthogonal experiment design.

Group Number	Moisture Content (%)	Concentration of CaCl_2_ (mol/L)	Ratio of CaCl_2_ to Urea	Empty Column	Empty Column	Empty Column
1	(1) 13%	(1) 2.0	(1) 1:2	(1) -	(1) -	(1) -
2	(1) 13%	(2) 4.0	(3) 2:1	(4) -	(5) -	(2) -
3	(1) 13%	(3) 3.0	(5) 1:3	(2) -	(4) -	(3) -
4	(1) 13%	(4) 5.0	(2) 1:1	(5) -	(3) -	(4) -
5	(1) 13%	(5) 1.0	(4) 3:1	(3) -	(2) -	(5) -
6	(2) 10%	(1) 2.0	(5) 2:1	(4) -	(3) -	(5) -
7	(2) 10%	(2) 4.0	(2) 1:3	(2) -	(2) -	(1) -
8	(2) 10%	(3) 3.0	(4) 1:1	(5) -	(1) -	(2) -
9	(2) 10%	(4) 5.0	(1) 3:1	(3) -	(5) -	(3) -
10	(2) 10%	(5) 1.0	(3) 1:2	(1) -	(4) -	(4) -
11	(3) 11%	(1) 2.0	(4) 1:3	(2) -	(5) -	(4) -
12	(3) 11%	(2) 4.0	(1) 1:1	(5) -	(4) -	(5) -
13	(3) 11%	(3) 3.0	(3) 3:1	(3) -	(3) -	(1) -
14	(3) 11%	(4) 5.0	(5) 1:2	(1) -	(2) -	(2) -
15	(3) 11%	(5) 1.0	(2) 2:1	(4) -	(1) -	(3) -
16	(4) 9%	(1) 2.0	(3) 1:1	(5) -	(2) -	(3) -
17	(4) 9%	(2) 4.0	(5) 3:1	(3) -	(1) -	(4) -
18	(4) 9%	(3) 3.0	(2) 1:2	(1) -	(5) -	(5) -
19	(4) 9%	(4) 5.0	(4) 2:1	(4) -	(4) -	(1) -
20	(4) 9%	(5) 1.0	(1) 1:3	(2) -	(3) -	(2) -
21	(5) 12%	(1) 2.0	(2) 3:1	(3) -	(4) -	(2) -
22	(5) 12%	(2)4.0	(4) 1:2	(1) -	(3) -	(3) -
23	(5) 12%	(3) 3.0	(1) 2:1	(4) -	(2) -	(4) -
24	(5) 12%	(4) 5.0	(3) 1:3	(2) -	(1) -	(5) -
25	(5) 12%	(5) 1.0	(5) 1:1	(5) -	(5) -	(1) -

**Table 3 materials-15-06529-t003:** The orthogonal experiment results.

Group Number	Average UCS (MPa)	Group Number	Average UCS (MPa)
1	3.43	14	5.92
2	5.22	15	2.46
3	6.02	16	8.89
4	9.31	17	12.14
5	4.32	18	12.92
6	5.43	19	9.42
7	8.62	20	7.65
8	9.88	21	4.90
9	6.56	22	6.04
10	4.32	23	10.20
11	5.15	24	3.25
12	8.11	25	4.06
13	6.97		

**Table 4 materials-15-06529-t004:** The orthogonal experiment analysis table.

Group Number	Moisture Content	Concentration of CaCl_2_(mol/L)	Ratio of CaCl_2_ to Urea
K_1_	(13%) 28.30	(2.0) 27.80	(1:2) 35.95
K_2_	(10%) 34.81	(4.0) 40.13	(1:1) 38.21
K_3_	(11%) 28.61	(3.0) 45.99	(2:1) 28.65
K_4_	(9%) 51.02	(5.0) 34.46	(3:1) 34.81
K_5_	(12%) 28.45	(1.0) 22.81	(1:3) 33.57
R	22.72	23.18	9.56

**Table 5 materials-15-06529-t005:** The orthogonal experiment result analysis. * Significance of Influence.

Source of Variation	Sum of Squares of Separation	Degree of Freedom	Mean Square	F-Value
Moisture content	76.48	4	19.12	7.65 (*)
Concentration of CaCl_2_	68.98	4	17.25	6.90 (*)
Ratio of CaCl_2_ to urea	10.14	4	2.54	1.01
Errors	29.99	12	2.50	

**Table 6 materials-15-06529-t006:** The calculating strategy of CaCO_3_ distribution characteristic.

Direction		Lateral	
	Location	Edge	Middle
Vertical	Upside	1	2
	Middle	4	5
	Downside	7	8

## Data Availability

All the data in the tests of this study have been listed in the paper.

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
