# Peer review of "Incorporation of Mixing Microbial Induced Calcite Precipitation (MICP) with Pretreatment Procedure for Road Soil Subgrade Stabilization"

_materials, 2022, doi:10.3390/ma15196529_

Round 1
Reviewer 1 Report
Current reference is used in the article. However, some articles on similar topics can also be added.
Similar studies on the use of bacteria should be added.
Author Response
Response to Reviewer 1 Comments
Point 1. Current reference is used in the article. However, some articles on similar topics can also be added.
Answer (A): Thank you for your comments. The articles on similar topics have been added, here are the added articles:
Gowthaman, S.; Nakashima, K.; Kawasaki, S. A state-of-the-art review on soil reinforcement technology using natural plant fiber materials: Past findings, present trends and future directions. Materials 2018, 11, 553.
Farah, R.E.; Nalbantoglu, Z. Behavior of Geotextile-Encased Single Stone Column in Soft Soils. Arab. J. Sci. Eng. 2020, 45, 3877-3890, doi:10.1007/s13369-019-04299-3.
Lu, Y.; Liu, S.; Zhang, Y.; Li, Z.; Xu, L. Freeze-thaw performance of a cement-treated expansive soil. Cold Regions Science and Technology 2020, 170, 102926.
Mahasenan, N.; Smith, S.; Humphreys, K. The cement industry and global climate change: current and potential future cement industry CO2 emissions. In Proceedings of Greenhouse gas control technologies-6th international conference; pp. 995-1000.
Worrell, E.; Price, L.; Martin, N.; Hendriks, C.; Meida, L.O. Carbon dioxide emissions from the global cement industry. Annual review of energy and the environment 2001, 26, 303-329.
Point 2. Similar studies on the use of bacteria should be added.
Answer (A): Thank you for your comments. The articles on similar topics have been added, here are the added articles:
Mwandira, W.; Nakashima, K.; Kawasaki, S. Bioremediation of lead-contaminated mine waste by Pararhodobacter sp. based on the microbially induced calcium carbonate precipitation technique and its effects on strength of coarse and fine grained sand. Ecological engineering 2017, 109, 57-64, doi:10.1016/j.ecoleng.2017.09.011.
Peng, S.; Di, H.; Fan, L.; Fan, W.; Qin, L. Factors affecting permeability reduction of MICP for fractured rock. Frontiers in Earth Science 2020, 10.3389/feart.2020.00217, 217, doi:10.3389/feart.2020.00217.
Liu, K.-W.; Jiang, N.-J.; Qin, J.-D.; Wang, Y.-J.; Tang, C.-S.; Han, X.-L. An experimental study of mitigating coastal sand dune erosion by microbial-and enzymatic-induced carbonate precipitation. Acta Geotechnica 2021, 16, 467-480, doi:10.1007/s11440-020-01046-z.
Xiao, Y.; Ma, G.; Wu, H.; Lu, H.; Zaman, M. Rainfall-Induced Erosion of Biocemented Graded Slopes. International Journal of Geomechanics 2022, 22, 04021256, doi:10.1061/(ASCE)GM.1943-5622.0002239.
Liu, S.; Wang, R.; Yu, J.; Peng, X.; Cai, Y.; Tu, B. Effectiveness of the anti-erosion of an MICP coating on the surfaces of ancient clay roof tiles. Construction and Building Materials 2020, 243, 118202, doi:10.1016/j.conbuildmat.2020.118202.
Zamani, A.; Montoya, B. Undrained monotonic shear response of MICP-treated silty sands. Journal of Geotechnical and Geoenvironmental Engineering 2018, 144, 04018029, doi:10.1061/(ASCE)GT.1943-5606.0001861.

Reviewer 2 Report
Edits:
1. Double full stop needs to be removed from the end of the sentence on line 21.
2. On line 31, "...there is a continually..." should read "...there has been a continually...", as the the sentence is referring to events in the past tense ("...during the past decades.")
3. Should "road structure" on line 36 be plural, "road structures"?
4. The term "situ" is used multiple times, with the first example being on line 40. This term ought to read "in situ" (possibly written in italics as it is a Latin term?) in every case from line 40 onwards.
5. Can the term "low efficiency" be better replaced with the term "inefficient", on line 46?
6. On line 62, "attentions" should read the singular "attention".
7. On line 64, " Sporosarcina pasteurii" is used for the first time. As this is a Linnaean taxonomic classification, it should be written in italics. It is then also usual to change all subsequent referrals to S. pasteurii.
7. On line 64, "...widely lived..." (a poor phrase) could be replaced with "widely found" or the more simple phrase "abundant".
8. On line 65, "...becomes..." should be replaced with "...has become...".
9. Line 66; this whole sentence should be redrafted (most likely into multiple sentences), in order to 1) refer to each of the numbered equations, and 2) marry each of these references with a brief comment that explains what each equation is showing, and the importance of why they've been included, perhaps. As things stand, the equations have been dumped into the paper with very little effort made to explain their significance/relevance.
10. Line 74, "...pore throat..." should be plural "...pore throats...".
11. Line 77 - 78, "...enhancing the coarse-grained soil,..." should read "...enhancing coarse-grained soils,...", and the same correction should be applied to "the fine-grained soil...".
12. Line 88, the use of "...high activity bacterial..." is confusing. Should this be "...high bacterial activity...", or less likely "...high activity bacteria..."? Not sure what the second option would even mean, so would recommend the first correction option.
13. Line 90, "...calcium ion..." should be plural "...calcium ions...".
14. Line 99, "...orthogonal experiment was..." should be "...orthogonal experimentation was employed to obtain...".
15. Line 101, "Moreover..." does not make sense, here. "Next..." or "Following this..." appears to be more appropriate.
16. Line 106, the use of "Furthermore..." negates the "...as well." at the end of the sentence, which should be removed.
17. Line 101; this sentence should be redrafted. Suggest:
"Particles less than 0.075mm were analyzed by a laser particle sizer (Malvern Mastersizer 2000), while particles greater than 0.075mm were analyzed by the water washing method according to the Chinese Specification JTG 113 3430-2020."
18. Line 139, "...dried till constant weight at 105ºC." should be "...dried at 105ºC until it achieved a constant weight".
19. Line 142, "For the UCS test, the cylindrical specimens..." should be "For the UCS test, cylindrical specimens...".
20. Line 149, "...curing time..." should be plural "...curing times...".
21. Line 152; see comment 19.
22. Line 153; "Each layer should be compacted well..." should be "Each layer was compacted well...".
23. Line 165, the word "respectively" should be removed as it doesn't refer to any respective items in the sentence.
24. Line 188; this sentence ("The samples obtained from the cyclic specimen were needed to smooth the surface and dried at 105°C for 12h before test.") does not make clear sense. Is "cyclic" supposed to be "cylindrical", perhaps? The sentence currently suggests that it is the samples themselves that are needed in order to "smooth the surface". An interpretation of this sentence (after comparison to the content of a sentence found at line 193) leads to the following possible re-draft option:
"The samples obtained from the cylindrical specimens had their surfaces smoothed before they were then dried at 105°C for 12h before testing."
25. Line 196, "...orthogonal experiment..." should be "...orthogonal experimentation..."
26. Table 2 appears to have three columns titled "Empty column". Is this supposed to be the case?
27. Line 216, "...5.2 times than the No.15 group..." should be "...5.2 times more than the No.15 group...".
28. Line 230, see comment 13.
29. Line 248, see comment 23.
30. Line 274; whole sentence needs redrafting. Suggest:
"The average UCS of the PMMRS group was almost three times greater than the pure CLS and was 78% greater than the SMMRS group."
31. Line 287, "...there were difference in the shear response..." should be "...there were differences in the shear responses...".
32. Line 291, "...as the strain softening." should be "...as strain softening."
33. Line 304, starting a new paragraph with "Moreover" is inappropriate and the sentence contains typographical/grammatical errors. Suggest:
"The shear strength envelope for the...".
34. Line 307; consistency needed. Is it "33.98 kPa" (with space) or "21.57kPa" (without space).
35. Line 311, "...which increased..." should be "...which increases...". Also, "...could also..." should be "...may...".
36. Line 312, "Compare..." should be "Compared...".
37. Line 325, delete one of either "detailed" or "described".
38. Line 330, starting a new paragraph with "Moreover" is inappropriate. Suggest:
"The average CCC for each specimen was used...".
39. Line 338, "...increased by 74% than the SMMRS..." should be "...increased by 74% more than the SMMRS...".
40. Line 340, "...could provide two times of calcium ion for mineralized..." should be "...could provide two times the calcium ion amount for mineralized...".
41. Line 365, "...oxygen existed in the pore..." should be "...oxygen existing in the pore...".
42. Line 366, "...as far away from the surface." could be "...if far away from the surface." or "...when far away from the surface.".
43. Line 367, "...in solution and pore were consumed completely..." should be "...in solution and in the pores is consumed completely...".
44. Line 375; sentence needs redrafting. Suggest:
"After the specimens were dried at 105°C for 24hrs, they were subjected to UCS testing.".
45. Line 404, "...imagines..." should be "...images...".
46. Line 432, "...the proportions of pore with little size..." should be "...the proportion of pores with a smaller size...".
47. Line 434; sentence needs redrafting. Suggest:
"Since the small soil particles could be bonded together, the amount of little particles would decrease, leading the soil pores that are normally filled with these little particles to become unblocked.".
48. Line 442, "...project..." should be plural "...projects...".
49. Line 458. The whole first sentence didn't make sense, and could actually just be deleted.
50. Line 463, "...was not so evenly..." should be "...was not even...".
51. Line 472, "...for making the CaCO3 generated evenly..." should be "...for ensuring the CaCO3 is generated more evenly...".
52. Line 47, "...long-term performance of PMMRS is also needed to be investigated..." should be "...long-term performance of PMMRS should also be investigated...".
Author Response
Response to Reviewer 2 Comments
Point 1. Double full stop needs to be removed from the end of the sentence on line 21.
Answer (A): Thank you for your comments. The double full stop has been removed from the end of the sentence on line 21.
Point 2. On line 31, "...there is a continually..." should read "...there has been a continually...", as the sentence is referring to events in the past tense ("...during the past decades.")
Answer (A): Thank you for your comments. The sentence on line 31 “there is a continually increasing demand for road construction in China during the past decades.” has been corrected as “there has been a continually increasing demand for road construction in China during the past decades.”
Point 3. Should "road structure" on line 36 be plural, "road structures"?
Answer (A): Thank you for your comments. The term "road structure" on line 36 has been corrected as plural, "road structures".
Point 4. The term "situ" is used multiple times, with the first example being on line 40. This term ought to read "in situ" (possibly written in italics as it is a Latin term?) in every case from line 40 onwards.
Answer (A): Thank you for your comments. The term “situ” has been corrected as the term “in situ”. The whole manuscript has been carefully re-checked and corrected.
Point 5. Can the term "low efficiency" be better replaced with the term "inefficient", on line 46?
Answer (A): Thank you for your comments. The “low efficiency" has been replaced with the term “inefficient” on line 47.
Point 6. On line 62, "attentions" should read the singular "attention".
Answer (A): Thank you for your comments. The “attentions" has been corrected as singular "attention".
Point 7. On line 64, " Sporosarcina pasteurii" is used for the first time. As this is a Linnaean taxonomic classification, it should be written in italics. It is then also usual to change all subsequent referrals to S. pasteurii.
Answer (A): Thank you for your comments. The “Sporosarcina pasteurii" on line 64 has been written in italics. And the “Sporosarcina pasteurii" has been corrected as “S. pasteurii” on line 119, 357, 360 and 363.
Point 8. On line 64, "...widely lived..." (a poor phrase) could be replaced with "widely found" or the more simple phrase "abundant".
Answer (A): Thank you for your comments. The sentence on line 64 has been corrected as “Moreover, Sporosarcina pasteurii, which is widely found in natural soil”.
Point 9. On line 65, "...becomes..." should be replaced with "...has become...".
Answer (A): Thank you for your comments. The sentence on line 65 has been corrected as “Moreover, Sporosarcina pasteurii, which is widely found in natural soil, has high urease activity and has becomes one of the most popular MICP bacteria specie nowadays.” in revised manuscript.
Point 10. Line 66; this whole sentence should be redrafted (most likely into multiple sentences), in order to 1) refer to each of the numbered equations, and 2) marry each of these references with a brief comment that explains what each equation is showing, and the importance of why they've been included, perhaps. As things stand, the equations have been dumped into the paper with very little effort made to explain their significance/relevance.
Answer (A): Thank you for your comments. The sentences have been redrafted into multiple sentences and added more information for more precise expression in revised manuscript. And here is the redrafted part:
The main chemical reactions of urea hydrolysis MICP are summarized as equation 1 to 6. And the mineralization reaction of S. pasteurii is often considered as being catalyzed by the carbonic anhydrase and urease. While the urease could hydrolyze the urea into ammonia and carbon dioxide, thus rapidly increasing the pH value and carbonate concentration of the cell microenvironment (equation 1, 2, and 3). And the carbonic anhydrase could catalyze the hydration reaction of carbon dioxide, forming carbonate and bicarbonate (equation 4 and 5), thus generating CaCO3 (equation 6).
|
CO(NH2)2+H2O→NH3+NH2COOH |
(1) |
|
NH2COOH+H2O→H2CO3+NH3 |
(2) |
|
NH3+H2O↔NH4++OH– |
(3) |
|
H2CO3↔HCO3–+H+ |
(4) |
|
HCO3–+H++2OH-↔CO32-+2H2O |
(5) |
|
Ca2++ CO3–+HCO3–+OH–→2CaCO3+H2O |
(6) |
Point 11. Line 74, "...pore throat..." should be plural "...pore throats...".
Answer (A): Thank you for your comments. The “pore throat” has been corrected as plural “pore throats” in revised manuscript.
Point 12. Line 77 - 78, "...enhancing the coarse-grained soil,..." should read "...enhancing coarse-grained soils,...", and the same correction should be applied to "the fine-grained soil...".
Answer (A): Thank you for your comments. The “the coarse-grained soil” and has been corrected as “the coarse-grained soils” on line 77, and “the fine-grained soil” has been corrected as “the fine-grained soils” on line 78 and 81, respectively.
Point 13. Line 88, the use of "...high activity bacterial..." is confusing. Should this be "...high bacterial activity...", or less likely "...high activity bacteria..."? Not sure what the second option would even mean, so would recommend the first correction option.
Answer (A): Thank you for your comments. The “the solution with high concentration and high activity bacterial” has been corrected as “the solution with high concentration and high bacterial activity” on line 89.
Point 14. Line 90, "...calcium ion..." should be plural "...calcium ions...".
Answer (A): Thank you for your comments. The “calcium ion” has been corrected as be plural "calcium ions" on line 90.
Point 15. Line 99, "...orthogonal experiment was..." should be "...orthogonal experimentation was employed to obtain...".
Answer (A): Thank you for your comments. The sentence “Firstly, orthogonal experiment was employed to obtain” on line 99 has been corrected as “Firstly, orthogonal experimentation was employed to obtain”.
Point 16. Line 101, "Moreover..." does not make sense, here.” Next..." or "Following this..." appears to be more appropriate.
Answer (A): Thank you for your comments. The word “Moreover” has been corrected as “Next” on line 101.
Point 17. Line 106, the use of "Furthermore..." negates the "...as well." at the end of the sentence, which should be removed.
Answer (A): Thank you for your comments. “Furthermore” has been removed in the revised manuscript. The sentence has been corrected as “And the PMMRS was studied by the field emission scanning electron microscope (FESEM), X-Ray Diffraction (XRD) and mercury injection apparatus (MIA) as well.”
Point 18. Line 101; this sentence should be redrafted. Suggest:
"Particles less than 0.075mm were analyzed by a laser particle sizer (Malvern Mastersizer 2000), while particles greater than 0.075mm were analyzed by the water washing method according to the Chinese Specification JTG 113 3430-2020."
Answer (A): Thank you for your comments. The sentence has been redrafted as suggested in the revised manuscript.
Point 19. Line 139, "...dried till constant weight at 105ºC." should be "...dried at 105ºC until it achieved a constant weight".
Answer (A): Thank you for your comments. The sentence has been corrected as “Finally, the mixed CLS was dried at 105°C until it achieved a constant weight.” in the revised manuscript.
Point 20. Line 142, "For the UCS test, the cylindrical specimens..." should be "For the UCS test, cylindrical specimens...".
Answer (A): Thank you for your comments. The sentence has been corrected as “For the UCS test, cylindrical specimens with a diameter of 50mm and a height of 50mm were prepared.” in the revised manuscript.
Point 21. Line 149, "...curing time..." should be plural "...curing times...".
Answer (A): Thank you for your comments. The “curing time” has been corrected as “curing times” in the revised manuscript.
Point 22. Line 152; see comment 19.
Answer (A): Thank you for your comments. The sentence has been corrected as “For the UU test, specimens with a diameter of 39.1 mm and a height of 80 mm were compacted with a hammer in four layers.” in the revised manuscript.
Point 23. Line 153; "Each layer should be compacted well..." should be "Each layer was compacted well...".
Answer (A): Thank you for your comments. The sentence has been corrected as “Each layer was compacted well before adding CLS mixture for the next layer.” in the revised manuscript.
Point 24. Line 165, the word "respectively" should be removed as it doesn't refer to any respective items in the sentence.
Answer (A): Thank you for your comments. The word "respectively" has removed in revised manuscript.
Point 25. Line 188; this sentence ("The samples obtained from the cyclic specimen were needed to smooth the surface and dried at 105°C for 12h before test.") does not make clear sense. Is "cyclic" supposed to be "cylindrical", perhaps? The sentence currently suggests that it is the samples themselves that are needed in order to "smooth the surface". An interpretation of this sentence (after comparison to the content of a sentence found at line 193) leads to the following possible re-draft option:
"The samples obtained from the cylindrical specimens had their surfaces smoothed before they were then dried at 105°C for 12h before testing."
Answer (A): Thank you for your comments. The sentence has been redrafted as “The samples obtained from the cylindrical specimens had their surfaces smoothed before they were then dried at 105°C for 12h before testing” in revised manuscript.
Point 26. Line 196, "...orthogonal experiment..." should be "...orthogonal experimentation..."
Answer (A): Thank you for your comments. The “orthogonal experiment” has been corrected as “orthogonal experimentation” on line 196.
Point 27. Table 2 appears to have three columns titled "Empty column". Is this supposed to be the case?
Answer (A): Thank you for your comments. In this study, since there were 3 factors within 5 levels, the orthogonal experimentation table L25 (56) was chosen, which has 6 columns and 25 levels. And because there were only 3 factors in this study, only the former 3 columns were used in this study, the rest 3 column would not be used in this study, thus they would be “empty columns”. But to reveal the truly orthogonal experimentation design plan, we chose to put the whole orthogonal experimentation table including these "Empty columns" into the manuscript.
Point 28. Line 216, "...5.2 times than the No.15 group..." should be "...5.2 times more than the No.15 group...".
Answer (A): Thank you for your comments. The sentence has been corrected as “which was 5.2 times more than the No.15 group” on line 216.
Point 29. Line 230, see comment 13.
Answer (A): Thank you for your comments. The phrase “calcium ion” has been corrected as be plural "calcium ions" on line 230.
Point 30. Line 248, see comment 23.
Answer (A): Thank you for your comments. The "respectively" has removed on line 248.
Point 31. Line 274; whole sentence needs redrafting. Suggest:
"The average UCS of the PMMRS group was almost three times greater than the pure CLS and was 78% greater than the SMMRS group."
Answer (A): Thank you for your comments. The sentence has been redrafted as suggest in revised manuscript on line 274.
Point 32. Line 287, "...there were difference in the shear response..." should be "...there were differences in the shear responses...".
Answer (A): Thank you for your comments. The sentence has been corrected as “there were difference in the shear responses among the PMMRS, SMMRS and pure CLS specimens.” in revised manuscript.
Point 33. Line 291, "...as the strain softening." should be "...as strain softening."
Answer (A): Thank you for your comments. The phrase “as the strain softening” has been corrected as “as strain softening” on line 291.
Point 34. Line 304, starting a new paragraph with "Moreover" is inappropriate and the sentence contains typographical/grammatical errors. Suggest:
"The shear strength envelope for the...".
Answer (A): Thank you for your comments. The sentence has been corrected as suggested: “The Shear strength envelope for the PMMRS, SMMRS and pure CLS were obtained and the results are shown in Figure 6.”
Point 35. Line 307; consistency needed. Is it "33.98 kPa" (with space) or "21.57kPa" (without space).
Answer (A): Thank you for your comments. The “21.57kPa" has been corrected as “21.57 kPa" on line 307.
Point 36. Line 311, "...which increased..." should be "...which increases...". Also, "...could also..." should be "...may...".
Answer (A): Thank you for your comments. The sentences have been corrected as “which increases the friction angle of the soil” and “the CaCO3 crystals may bond soil particles together and hence improve the cohesion of the soil” in revised manuscript, respectively.
Point 37. Line 312, "Compare..." should be "Compared...".
Answer (A): Thank you for your comments. The word has been corrected as “Compared” on line 312.
Point 38. Line 325, delete one of either "detailed" or "described".
Answer (A): Thank you for your comments. The "detailed" has been deleted on line 325 in revised manuscript.
Point 39. Line 330, starting a new paragraph with "Moreover" is inappropriate. Suggest:
"The average CCC for each specimen was used...".
Answer (A): Thank you for your comments. The sentences have been corrected as “The average CCC for the specimen was used to study the relationship between the CCC in specimens and the CaCl2 concentration in cementation solution.” in revised manuscript.
Point 40. Line 338, "...increased by 74% than the SMMRS..." should be "...increased by 74% more than the SMMRS...".
Answer (A): Thank you for your comments. The sentences have been corrected as “Figure 8 illustrates that the average CCC of PMMRS was increased by 74% more than the SMMRS while SMMRS had a higher CICR.” in revised manuscript.
Point 41. Line 340, "...could provide two times of calcium ion for mineralized..." should be "...could provide two times the calcium ion amount for mineralized...".
Answer (A): Thank you for your comments. The sentences have been corrected as “pretreatment-mixing MICP method could provide two times of calcium ion for mineralized bacteria to generate CaCO3 theoretically” in revised manuscript.
Point 42. Line 365, "...oxygen existed in the pore..." should be "...oxygen existing in the pore...".
Answer (A): Thank you for your comments. The sentences have been corrected as “including the dissolved oxygen in solution, oxygen existed existing in the pore” in revised manuscript.
Point 43. Line 366, "...as far away from the surface." could be "...if far away from the surface." or "...when far away from the surface.".
Answer (A): Thank you for your comments. The sentences have been corrected as “it is hard for the bacteria inside the specimens to obtain the oxygen from the environment as if far away from the surface.” in revised manuscript.
Point 44. Line 367, "...in solution and pore were consumed completely..." should be "...in solution and in the pores is consumed completely...".
Answer (A): Thank you for your comments. The sentences have been corrected as “once the oxygen in solution and in the pores were consumed completely” in revised manuscript.
Point 45. Line 375; sentence needs redrafting. Suggest:
"After the specimens were dried at 105°C for 24hrs, they were subjected to UCS testing.".
Answer (A): Thank you for your comments. The sentences have been redrafted as suggested on line 375.
Point 46. Line 404, "...imagines..." should be "...images...".
Answer (A): Thank you for your comments. The word has been corrected as “images” on line 404.
Point 47. Line 432, "...the proportions of pore with little size..." should be "...the proportion of pores with a smaller size...".
Answer (A): Thank you for your comments. The sentences have been corrected as “the proportion of pores with a smaller size for all the MICP reinforced soil were smaller than the pure CLS.” in revised manuscript.
Point 48. Line 434; sentence needs redrafting. Suggest:
"Since the small soil particles could be bonded together, the amount of little particles would decrease, leading the soil pores that are normally filled with these little particles to become unblocked.".
Answer (A): Thank you for your comments. The sentences have been redrafted as suggested on line 434 in revised manuscript.
Point 49. Line 442, "...project..." should be plural "...projects...".
Answer (A): Thank you for your comments. The word has been corrected as “projects” on line 404.
Point 50. Line 458. The whole first sentence didn't make sense, and could actually just be deleted.
Answer (A): Thank you for your comments. The sentence has been deleted on line 458.
Point 51. Line 463, "...was not so evenly..." should be "...was not even...".
Answer (A): Thank you for your comments. The phrase has been corrected as “was not even” on line 463.
Point 52. Line 472, "...for making the CaCO3 generated evenly..." should be "...for ensuring the CaCO3 is generated more evenly...".
Answer (A): Thank you for your comments. The sentences have been corrected as “it is strongly recommended to investigate the methodology for ensuring the CaCO3 is generated more evenly in the PMMRS.”
Point 53. Line 47, "...long-term performance of PMMRS is also needed to be investigated..." should be "...long-term performance of PMMRS should also be investigated...".
Answer (A): Thank you for your comments. The sentences have been corrected as “Furthermore, long-term performance of PMMRS is should also be needed to be investigated in order to verify the indoor test results in further studies.”

Reviewer 3 Report
Dear Editor,
I include herein comments in relation to the review of Manuscript materials-1865573, entitled “Incorporation of Microbial Induced Calcite Precipitation (MICP) with Pretreatment Procedure for Road Soil Subgrade Stabilization. The research is of great interest and the objectives, methodology and results are clearly explained. There are a few issues to take into consideration but these need minor changes.
My recommendation is to publish it with minor changes.
Comments are included in the test.
My very best regards

Author Response
Response to Reviewer 3 Comments
Point 1. Line 17: Previously it has been indicated a series of parameters that have been analyzed and now the results indicate they depend on other variables. Please, indicate previously in the abstract that the CaCl2 and the urea have been other parameter taken into account in the analysis.
Answer (A): Thank you for your comments. In this study, the preparation parameters of the MICP reinforced soil including 3 factors: the dry density and moisture content of the soil (these two factors are directly linked under the standard compact work, so they could be considered as one factor), the concentration of CaCl2 and urea in the cementation solution, they determined the strength of the MICP reinforced soil, and they are all taken into account in the analysis and the orthogonal experimentation. Therefore, the abstract has been modified to express that the CaCl2 and urea have been taken into account in the analysis. Here is the correction part in revised manuscript: “A series of laboratory tests were performed to investigate the preparation parameters (including the moisture content and dry density of soil, the concentration of urea and CaCl2), the engineering properties, the CaCO3 distribution as well as the mineralogical and micro structural characteristics of pretreatment-mixing MICP reinforced soil (PMMRS).”
Point 2. Line 21: The double full stop needs to be removed
Answer (A): Thank you for your comments. The Double full stop has been removed from the end of the sentence on line 21.
Point 3. Line23: This needs to be explained, the oxygen content is not a reason to precipitate one polymorph or another.
Answer (A): Thank you for your comments. We agree with you that the oxygen content is not a reason to precipitate the polymorph of CaCO3, but the oxygen content could influence the growth and survival of Sporosarcina pasteurii, which is a kind of aerobic bacteria. And it has been testified by Mingdong Li[1] that the oxygen content could affect the reinforcement effect of MICP. Therefore, we thought that the oxygen content might be the reason for the uneven distribution of CaCO3 precipitation, and this will be a key point for our research in future. And previously, we wanted to express that “the oxygen content might be the reason for the uneven distribution of CaCO3” in the original manuscript, but we seemed to have inaccurate expression, leading your misunderstanding. The sentence has been corrected as “The reason for the uneven distribution might be that oxygen content varied with the regions in different directions” on line 23 in revised manuscript.
1.Li, M.; Wen, K.; Li, Y.; Zhu, L. Impact of oxygen availability on microbially induced calcite precipitation (MICP) treatment. Geomicrobiology Journal 2018, 35, 15-22, doi:10.1080/01490451.2017.1303553.
Point 4. Line 25: “…reinforcement and then reinforced soil.” the same word is repeated. Look for a different one.
Answer (A): Thank you for your comments. We have used “stabilization” to replace “reinforcement”, and the sentence has been corrected as “which was highly related to the reinforcement stabilization effect of MICP reinforced soil” on line 26 in revised manuscript.
Point 5. Line 31: Should to be “has been an increasing demand for road construction in China”
Answer (A): Thank you for your comments. The sentence has been corrected as “there has been an increasing demand for road construction in China during the past decades.” on line 31 in revised manuscript.
Point 6. Line 33: removed “in China”
Answer (A): Thank you for your comments. The “in China” has been deleted, and the sentence has been corrected as “By the end of 2020, the total road mileage reached to 5,198,100 km.” in revised manuscript.
Point 7. Line 34: “…with…” should be “…between…”
Answer (A): Thank you for your comments. The sentence has been corrected as “the geological conditions might vary significantly between regions”.
Point 8. Line 37: add “character”
Answer (A): Thank you for your comments. The word “character” has been added, and the sentence has been corrected as “effective methods are needed to eliminate the potential risk caused by the poor soil character.” in revised manuscript.
Point 9. Line 40: add “developing”
Answer (A): Thank you for your comments. The word “developing” has been added, and the sentence has been corrected as “(b) developing physical engineering methods”.
Point 10. Line 43 add “producing”
Answer (A): Thank you for your comments. The word “producing” has been added, and the sentence has been corrected as “(d) producing inorganic binder stabilization” in revised manuscript.
Point 11. Line 43: “Poland cement” should be “Portland cement”
Answer (A): Thank you for your comments. The word “Poland cement” has been corrected as “Portland cement” in revised manuscript.
Point 12. Line 45: “…disadvantages that needed to be…” should be “…disadvantages that need to be…”
Answer (A): Thank you for your comments. The sentence has been corrected as “but has considerable disadvantages that needed to be seriously considered”.
Point 13. Line 47: “efficiency” should be “efficient”
Answer (A): Thank you for your comments. The “low efficiency" has been corrected as “inefficient” on line 47.
Point 14. Line 56: delete “so”
Answer (A): Thank you for your comments. The word “producing” has been added, and the sentence has been corrected as “the cement binder stabilization is not environmentally friendly since cement production consumes a large amount of energy and significantly increases CO2 emissions” in revised manuscript.
Point 15. Line 58: add “as a”
Answer (A): Thank you for your comments. The word “producing” has been added, and the sentence has been corrected as “and as a heavy metal pollution treatment” in revised manuscript.
Point 16. Line 61: delete “the”
Answer (A): Thank you for your comments. The word the” has been added, and the sentence has been corrected as “which is an emerging and eco-friendly technique based on the microbial mineralization” in revised manuscript.
Point 17. Line 62: delete “the”
Answer (A): Thank you for your comments. The word “the” has been added, and the sentence has been corrected as “has been developed and applied in soil reinforcement” in revised manuscript.
Point 18. Line 64: “… lived in…” should be “…distributed in natural soils…”
Answer (A): Thank you for your comments. The sentence on line 64 has been corrected as “Moreover, Sporosarcina pasteurii, which is widely found in natural soil” in revised manuscript.
Point 19. Line 70: “injecting method” should be “the injecting method”
Answer (A): Thank you for your comments. The word “injecting method” has been corrected as “the injecting method” on line 70 in revised manuscript.
Point 20. Line 70: “spraying method” should be “the spraying method”
Answer (A): Thank you for your comments. The word “spraying method” has been corrected as “the spraying method” on line 70 in revised manuscript.
Point 21. Line 70: “soaking method” should be “the soaking method”
Answer (A): Thank you for your comments. The word “soaking method” has been corrected as “the soaking method” on line 70 in revised manuscript.
Point 22. Line 71: “mixing method” should be “the soaking method”
Answer (A): Thank you for your comments. The word “mixing method” has been corrected as “the mixing method” on line 71 in revised manuscript.
Point 23. Line 72: “… are needed to diffuse from…” should be “… need to be diffused from…”
Answer (A): Thank you for your comments. The sentence on line 64 has been corrected as “the bacteria and cement substances need to be diffused from the surface to inside or from one side to the opposite.” in revised manuscript.
Point 24. Line 99: add “an”
Answer (A): Thank you for your comments. The word “an” has been added, and the sentence has been corrected as “Firstly, an orthogonal experimentation was employed to obtain the optimum parameters for preparing the pretreatment-mixing MICP reinforced soil (PMMRS).” in revised manuscript.
Point 25. Line 109: it is impossible to believe that all particles had a 0.075 mm size. This grain size is commonly gives within and interval (0,002 – 0,05 mm silt size or << 0,002mm clay size)
Answer (A): Thank you for your comments. In this study, the soil not only has the soil particles smaller than 0.075mm, but also has the soil particles greater than 0.075mm, not all of soil particles have a 0.075mm. The use of “the soil particle within 0.075mm” in this sentence is a mistake, we previously wanted to express that “the soil particle less than 0.075mm were analyzed by a laser particle sizer.” And this sentence has been corrected as “The particles less than 0.075mm were analyzed by a laser particle sizer” in revised manuscript.
Point 26. Line 113: please give grain size in intervals
Answer (A): Thank you for your comments. For the soil used in this study, about 49% of soil particles are smaller than 0.075mm (75μm), and about 51% of soil particles are greater than 0.075mm. Therefore, we think that the distribution of soil particles greater than 0.075mm is as important as the distribution of soil particles smaller than 0.075mm. And to reveal the distribution of soil particles in all grain size, we chose to use logarithmic axis but not isometric axis in the “Particle size distribution of the CLS” figure. If we give grain size in same intervals in the particle size distribution figure, the distribution of the soil particle smaller than 0.075mm will be nearly invisible. Consequently, we think it would be better to not give grain size in intervals in “Particle size distribution of the CLS” figure.
Point 27. Remove “Figure 1 presents the size distribution of the CLS.” and add (Fig.1)
Answer (A): Thank you for your comments. The sentence has been deleted and the “Figure.1” has been added to the former sentence. Here is the correction in revised manuscript: “while the particles greater than 0.075mm were analyzed by the water washing method according to the Chinese Specification JTG 3430-2020 (Figure.1) .”
Point 28. Line148: please write 300 ºC
Answer (A): Thank you for your comments. This is a mistake, we previously wanted to write 30 centigrade, we apologize for our carelessness. The sentence has been corrected as “the bacteria-bearing liquid culture medium was cultured in an oscillating incubator at 30ºC with a vibration rate of 180 rpm for 24 hours.”
Point 29. Line 168: “… in the PMMRS specimens was investigated …” should be “… in the PMMRS specimens were investigated …” (they are the content and distribution, and these are two parameters investigated)
Answer (A): Thank you for your comments. The sentence has been corrected as “The content and distribution of CaCO3 in the PMMRS specimens was were investigated by the acid washing method”.
Point 30. Line 196: add “an”
Answer (A): Thank you for your comments. The sentence has been corrected as “an orthogonal experimentation was used to investigate the different factors that influence the reinforcement effect of pretreatment-mixing MICP method”.
Point 31. Line 268: remove “recorded, and the results are shown in Figure 4.” and add (Fig.4)
Answer (A): Thank you for your comments. The sentence has been deleted and the “Figure.4” has been added to the former sentence. Here is the correction in revised manuscript: “During the UCS test process, both the stress applied to the specimens and the corresponding strain was recorded (Figure.4).”
Point 32. Line 272: Delete “As shown in Figure 4”, and add “The average”
Answer (A): Thank you for your comments. The sentence has been deleted and the “The average” has been added. The sentence has been corrected as “The average UCS for the PMMRS group was the greatest.”.
Point 33. Line 303: Write a final point at the end of all figure captions.
Answer (A): Thank you for your comments. A final point has been added at the end of all figure captions in revised manuscript. Such as: “Figure 5. Shear response of specimens at confining pressure of 100 kPa.”
Point 34. Line 305: Deleted “and the results are shown in Figure 6” and add “(Fig. 6)”
Answer (A): Thank you for your comments. The sentence has been deleted and the “Figure.6” has been added to the former sentence. Here is the correction in revised manuscript: “The Shear strength envelope for the PMMRS, SMMRS and pure CLS were obtained (Figure 6).
Point 35. Line 338: Deleted “Figure 8 illustrates that the” and add “average” and “(Fig. 8)”
Answer (A): Thank you for your comments. The sentence has been deleted and the “average” and “(Figure. 8)” has been added. Here is the correction in revised manuscript: “The average CCC of PMMRS was increased by 74% more than the SMMRS while SMMRS had a higher CICR (Figure.8).”
Point 36. Line 382: Deleted “As shown in Figure 10,” and add “(Fig. 10)”
Answer (A): Thank you for your comments. The sentence has been deleted and the “(Figure. 10)” has been added. Here is the correction in revised manuscript: “UCS of PMMRS increased within the whole curing period (Figure.10).”
